# Adults Suffering from Violence Are at Risk of Poor Prognosis in Taiwan, 2000–2015

**DOI:** 10.3390/healthcare10081470

**Published:** 2022-08-04

**Authors:** Li-Yun Fann, Iau-Jin Lin, Shi-Hao Huang, Bing-Long Wang, Yao-Ching Huang, Chia-Peng Yu, Chih-Chien Cheng, Chien-An Sun, Cheng-Wei Hsu, Wu-Chien Chien, Chieh-Hua Lu

**Affiliations:** 1Department of Nursing, Taipei City Hospital, Taipei 10341, Taiwan; 2Department of Nurse-Midwifery and Women Health, National Taipei University of Nursing and Health Sciences, Taipei 11220, Taiwan; 3Department of Medical Research, Tri-Service General Hospital, Taipei 11490, Taiwan; 4School of Public Health, National Defense Medical Center, Taipei 11490, Taiwan; 5Department of Chemical Engineering and Biotechnology, National Taipei University of Technology (Taipei Tech), Taipei 10608, Taiwan; 6Department of Mechanical Engineering, National Central University, Jhongli 32001, Taiwan; 7Department of Obstetrics/Gynecology, Taipei City Hospital, Taipei 10341, Taiwan; 8School of Medicine, College of Medicine, Fu Jen Catholic University, New Taipei City 242062, Taiwan; 9Department of Public Health, College of Medicine, Fu-Jen Catholic University, New Taipei City 242062, Taiwan; 10Big Data Center, College of Medicine, Fu-Jen Catholic University, New Taipei City 242062, Taiwan; 11Graduate Institute of Life Sciences, National Defense Medical Center, Taipei 11490, Taiwan; 12Taiwanese Injury Prevention and Safety Promotion Association (TIPSPA), Taipei 11490, Taiwan; 13Department of Internal Medicine, Division of Endocrinology and Metabolism, Tri-Service General Hospital, School of Medicine, National Defense Medical Center, Taipei 11490, Taiwan

**Keywords:** adult maltreatment, injury, psychiatric disorders, poor prognosis

## Abstract

Objective: To understand the risk of developing a poor prognosis in adulthood after violent injury in Taiwan. Methods: This study used the data of outpatients, from emergency departments, and from hospitalization of 2 million people under National Health Insurance from 2000 to 2015. The ICD-9 diagnostic code N-code was defined as the case of this study and was 995.8 (abused adult) or E-code was E960-E969 (homicide and intentional injury by others) The first violent injury of 18–64-year-old adults (the study group) was analyzed. Patients who had not suffered violent abuse were the control group. The groups were matched in a 1:4 ratio, and the paired variables were gender, age ±1 year, Charlson Comorbidity index (CCI) before exposure, and year of medical treatment. SAS 9.4 statistical software was used, and the Cox regression method was used for data analysis. Results: During the 15-year period, a total of 8726 people suffered from violence (34,904 controls). The incidences of common poor prognoses among the victims of violence were sleep disorder, anxiety, and depression, in 33.9%, 21.6%, and 13.2% of people, respectively. The risk (Adults, Overall) of developing Post-Traumatic Stress Disorder (PTSD), bipolar disorder, and manic disorder after being violently injured (average 9 years) was 34.86, 4.4, and 4.1 times higher than those who had not suffered violence (all *p* values < 0.01). The risk (Adults, Males) of developing PTSD, bipolar disorder, and manic disorder after being violently injured (average 9 years) was 30.0, 3.81, and 2.85 times higher, respectively, than those who had not suffered violence (all *p* values < 0.01). The risk (Adults, Females) of developing PTSD, manic disorder, and bipolar disorder after being violently injured (average 9 years) was 36.8, 6.71, and 5.65 times higher, respectively, than of those who had not suffered violence (all *p* values < 0.01). Conclusion: The risks of poor prognosis are higher in adults who have suffered violent abuse than in those who have not. Therefore, police, social workers, and medical personnel should pay attention to the mental state of victims of violence. They should aim to support prompt treatment, to avoid PTSD, bipolar disorder, manic disorder, etc.

## 1. Introduction

Child abuse and neglect can affect an individual’s physical and mental health in both direct and indirect ways [1,2]. Abuse in infancy and early childhood has been shown to negatively impact early brain development, which, in turn, leads to negative behavioral health outcomes in adolescence and adulthood [3,4]. The link between early abuse and adverse adult health outcomes is well established [5]. The immediate emotional effects of abuse and neglect, such as isolation, fear, and inability to trust, can translate into lifelong consequences, including poor mental and behavioral health outcomes and an increased risk of substance use disorders [6].

Interpersonal violence involves the deliberate use of force or power over another by an individual or a small group of people. Interpersonal violence may be physical, sexual, or psychological (also known as emotional violence) and may involve deprivation and neglect [7]. According to the “World Report on Violence and Health” by the World Health Organization (WHO) in 2002, interpersonal violence is divided into domestic partner violence (family member or partner relationship) and community violence (without kinship) according to the relationship between the perpetrator and the victim. Among them, the WHO uses Intimate Partner Violence (IPV) for the classification of violence between non-related adults in a personal relationship [8,9]. The scope of violence between partners includes the following: physical aggression, such as slapping, kicking, and hitting; mental abuse, such as coercion, contempt, and humiliation; forced sex and other forms of sexual coercion; and various forms of control, such as isolating victims from family and friends, monitoring their activities, and limiting their access to information and help [8,9].

A study of 654,356 victims of interpersonal and domestic violence in the United States in 2014 showed that the risk of death in hospital was 2.1 times higher for patients with trauma from domestic and IPV than for non-trauma patients from domestic and IPV (aOR: 2.31, *p* < 0.001). Injury patterns in patients diagnosed with interpersonal violence or domestic violence trauma showed the most common to be head injury (vs. non-domestic violence: 16.1% vs. 3.4%, *p* < 0.001) and multiple injuries (vs. Compared: 16.6% vs. 7.4%, *p* < 0.001) [10]. A Korean study from 2018 performed a retrospective analysis of preliminary interviews and medical records of 239 victims of domestic violence who attended the Sunflower Center and Emergency Center in Korea from 1 January 2016 to 31 December 2016, and showed that 191 (79.9%) suffered a physical attack, with a mean New Injury Severity Score of 2.93. Injury types included contusions, abrasions, lacerations, fractures, subconjunctival hemorrhages, perforated tympanic membranes, broken teeth, and subarachnoid hemorrhage. Most physical injuries occurred in the face, upper extremities, head, and neck, and 45 (95.7%) of the 47 victims, who were physically injured, had a mean severity score of 5.40 ± 4.15 (Park et al., 2018) [11].

Chen et al. (2010) searched nine literature databases from 1980 to 2008 and conducted a comprehensive analysis of 37 papers, including 17 case-control studies and 20 generational studies examining the relationship between sexual abuse and lifetime psychiatric diagnoses. Analysis showed that sexual abuse was significantly associated with lifetime diagnoses of anxiety, depression, eating disorders, post-traumatic stress disorder (PTSD), sleep disturbances, and suicide attempts, regardless of the victim’s gender or age at the time of the abuse. However, there was no statistically significant association between exposure to sexual abuse and lifetime schizophrenia or psychosomatic disorders [12]. A 2015 study in the capital of Gilan province, Iran, administered a general health questionnaire to 2091 married women, who were divided into battered women (*n* = 512, 24.5%) and non-battered women (*n* = 1579, 75.5%). The research results showed that the proportion of psychological aggression in intimate partner violence is greater than that of physical aggression, sexual coercion, or injury, and abused persons are more prone to psychosomatic disorders, anxiety/insomnia, social dysfunction, and depression. In addition, studies have also found that both psychological and sexual abuse are important predictors of mental health [13]. A cross-sectional survey of home health care patients (444 women) in a small Italian town found that only psychological abuse, without sexual or physical violence, was associated with impaired health. After adjusting for age, education, presence of children, marital and employment status, women who experienced partner violence were six times more likely to suffer from depression and physical distress, and four times more likely to use psychotropic substances [14]. Furthermore, there is a strong correlation between past and present experiences of interpersonal violence [15].

At present, longitudinal observational studies on the relationship between adult maltreatment and poor prognosis are limited. Therefore, we hypothesize that adults suffering from violence are at risk of poor prognosis (anxiety, depression, manic disorder, bipolar disorder, sleep disorder (SD), PTSD, acute stress disorder (ASD), and eating disorders (ED)). We used the National Health Insurance Research Database (NHIRD) of the Ministry of Health and Welfare to investigate whether adults who experienced violence are at risk of poor prognosis from 2000–2015 in Taiwan through a long-term follow-up approach.

## 2. Method

### 2.1. Data Sources

This study used the NHIRD to provide a representative National Health Insurance (NHI) 2000 cohort of two million underwriting sample archives (Longitudinal Health Insurance Research Database, LHID2000). As the research data source, the follow-up year was from 1 January 2000, to 31 December 2015, for a total of 16 years of outpatient and inpatient data. This study used the data from the year 2000 as data cleaning to exclude non-new cases. The violently abused adults in this study included 18 to 64-year-olds, according to the International Classification of Diseases, Ninth Revision, Clinical Modification (ICD-9 CM) N-code: 995.8 and the classification of external causes Code definition E-code: E960–E969.

The sampling source of the control group included unstricken patients, and the case-control matching method was used to establish the control group according to the matching ratio of 1:4. The matching conditions included: gender, age (±1 year), primary violence time (year and month) (the same medical treatment year and month for the control group), and CCI (Charlson Comorbidity Index, accumulated to the CCI before the first exposure). Afterwards, two groups of data were paired according to the above conditions, the same numbers were given to those with the same pairing conditions for statistical analysis of the paired data. The CCI used in this study was a version modified by Deyo in 1992 [16]. Injury type was in accordanceICD9 codes 800–989, 995.80–995.85 (Adult abuse). This study applied to the Ministry of Health and Welfare to use the NHIRD of Taiwan’s NHI. The data analyzed did not contain personally identifiable information. The Ethics Review Board of the Tri-Services General Hospital of the National Defense Medical Center (TSGHIRB No.: C202105014) approved this study, waiving personal written informed consent. The source of funding for the project was the in-hospital research project of the Tri-Service General Hospital of the National Defense Medical College: TDGH-B-111018.

### 2.2. Study Design

This study used a retrospectively matched cohort design from 1 January 2000 to 31 December 2015 and used the ICD-9-CM code 995.80 to represent adult abuse. Victims over the age of 18 were included in the adult abused cohort (*n* = 8726), and adults without experience of abuse were included in the comparison cohort (*n* = 34,904) matched for sex, age, CCI, and index date (1:4). Poor prognosis included anxiety, depression, manic disorder, bipolar disorder, SD, PTSD, ASD, and ED. Victims with a history of adult abuse or poor prognosis before 2000 were excluded.

Covariates in this study included gender, age, residential location (north, central, south or east), urbanization level (grades 1–4), level of medical facility (medical center, regional hospital, local hospital), and insurance premiums category (New Taiwan Dollars [NTD]; <18,000, 18,000–34,999, ≥35,000).

Urbanization level was classified according to the population density (person/km^2^), the proportion of the population with a college education or above (%), the proportion of the population over the age of 65 (%), the population classified as migrant workers by occupation (%), and the number of migrant workers per city or county. Number of physicians per 100,000 population was also included [17].

CCI score with 17 relevant comorbidity categories (based on ICD-9-CM codes) was used for this study. Each comorbidity category had an associated weight (from 1 to 6), and the sum of all weights yielded a patient’s single comorbidity score based on the adjusted risk of mortality or resource use. A score of zero indicated that no comorbidities were found [18]. Figure 1 shows the study design flow chart for this study.

### 2.3. Statistical Analysis

This study was analyzed using SAS 9.4 for Windows (SAS Institute, Cary, NC, USA) statistical software provided by the Academia Sinica Branch of the Data Welfare Center of the Ministry of Health and Welfare. The Chi-square test, logistic regression, and Cox regression were used. The descriptive data of the two groups were tested by the Generalized Estimating Equations (GEE) method.

## 3. Results

### 3.1. Characteristics of the Study

This study included a case group of 8726 abused adults and a matched control group of 34,904 non-abused adults. The characteristic differences between the two groups in this study are shown in Table 1. The average age of abused adults was 36.9 ± 12.0 years, and the proportion of males was 67.1%. The differences between the two groups in this study lay in the following variables: insurance premium, CCI, urbanization level, and level of care.

### 3.2. Risk of Mental Illness and Poor Prognosis According to Adult Maltreatment Exposure

Adult victims of violent abuse have significantly higher risks of poor prognosis than victims of non-violent abuse. Among them, the three adverse outcomes with the greatest risk (Adults, Overall) difference were PTSD (Hazard Ratio (HR) = 34.86), bipolar disorder (HR = 4.44), and manic disorder (HR = 4.10) (Table 2).

When divided by gender, adult male victims of violence have significantly higher risks of poor prognosis than non-violent victims. Among them, the three adverse outcomes with the largest risk (Adults, Males) difference were PTSD (HR = 30.0), bipolar disorder (HR = 3.81), and manic disorder (HR = 2.85) (Table 3).

Adult female victims of violent abuse have significantly higher risks of poor prognosis than victims of non-violent abuse. These victims had a higher risk for PTSD (HR = 36.8), manic disorder (HR = 6.71), and bipolar disorder (HR = 5.65) (Table 4).

## 4. Discussion

The results showed that the risk of poor prognosis in adults who have been subjected to violent abuse is significantly higher than in those who have been subjected only to non-violent abuse, which is similar to the results of foreign studies [19,20,21,22,23,24,25]. One study demonstrated a significant documented mental health burden associated with IPV in primary care at baseline and after exposure [20]. Clinicians must be aware of this association to reduce delays in psychiatric diagnosis and improve the management of psychological outcomes in this group of patients [20]. The study provides evidence that severe physical IPV poses a significant threat to the mental health of black women in the United States [22]. In our study, there were three adverse outcomes posing the greatest risk (Adults, Overall) difference: PTSD (HR = 34.86), bipolar disorder (HR = 4.44), and manic disorder (HR = 4.10). The three adverse outcomes with the largest risk (Adults, Males) difference were PTSD (HR = 30.0), bipolar disorder (HR = 3.81), and manic disorder (HR = 2.85) and those with higher risk for women were PTSD (HR = 36.8), manic disorder (HR = 6.71), and bipolar disorder (HR = 5.65). Adults subjected to violent abuse have higher risks of poor prognosis; in particular, a higher risk of developing PTSD, bipolar disorder, and manic disorder.

Many different types of traumas have been found to cause PTSD, bipolar disorder, and manic disorder [26,27,28,29]. These types, and the proportion of PTSD, bipolar disorder, and manic disorder cases they make up, include the following:
Sexual relationship violence—33% (e.g., rape, childhood sexual abuse, intimate partner violence).Interpersonal traumatic experiences—30% (ego, accidental death of a loved one, life-threatening illness in a child, other traumatic events in a loved one).Interpersonal violence—12% (e.g., childhood physical abuse or witnessing interpersonal violence, physical attack, or threat of violence).Exposure to organized violence—3% (ego, refugees, kidnappings, civilians in war zones).Participation in organized violence—11% (e.g., combat exposure, witnessing death/serious injury or finding a dead body, accidental or intentional death or serious injury).Other life-threatening traumatic events—11% (ego, life-threatening motor vehicle collisions, natural disasters, exposure to toxic chemicals).

The risk of poor prognosis for violently injured people was significantly higher than that for non-violently injured people [30]. Traumatic events, such as domestic and social violence, rape and assault, disasters, wars, accidents, and predatory violence, expose people to such terror and threats that it may, temporarily or permanently, alter their coping abilities, perceptions of biological threats, and their conception of themselves [31]. The human response to psychological trauma is one of the world’s most important public health problems [32]. For all injuries and violence, providing victims with high-quality emergency care can prevent deaths, reduce the number of short- and long-term disabilities, and illustrate how those affected are physically, emotionally, financially, and legally responding to the injury or violence impacting their lives [33]. Therefore, improving the organization, planning, and access to trauma care systems, including telecommunications, hospital transport, pre-hospital, and hospital care, are important strategies for minimizing death and disability due to injury and violence [34]. Several studies have explored the association of risk of poor prognosis for those who have experienced violence, but the number of study cases is small and the follow-up time is not long enough, so the association between the risk of poor prognosis for adults in Taiwan and suffering from violence remains unclear. The sample size of our study was large enough to represent Taiwan’s population of 23 million “adults exposed to violence at risk of poor prognosis” and to provide “violence injury” risk factors on various kinds of poor prognosis. Through NHIRD, we were able to confirm the relationship between “adults exposed to violence at risk of poor prognosis”. We found that violent injury had a significant risk of poor prognosis for the victims of violence, and, through precise matching, the differences between the victims and those who had not suffered violent abuse were minimized. Moreover, the results were similar in both males and females, reinforcing the researchers’ confidence in the consistency of the results.

The study has several limitations that warrant consideration. First, similar to the previous study using the NHI research database on psychiatric disorders [35], we were unable to evaluate genetic, psychosocial, or environmental factors, severity, or psychological assessments in the patients with psychiatric disorders, since these data were not recorded in the NHI research database. Second, the subjects of this study were adults who had been exposed to violence and sought medical care. Therefore, only those who have experienced more severe violence were observed, and the total incidence of violent abuse may be underestimated. Individuals who have suffered psychological abuse without obvious trauma, or those who did not seek medical treatment, due to fear, could remain unaccounted for.

## 5. Conclusions

The results of this follow-up study support a strong association between abused adults and an increased risk of PTSD, bipolar disorder, manic disorder, etc. Adults subjected to violence have a higher risk of poor prognosis, in particular of developing PTSD, bipolar disorder, and manic disorder. Through this study, it is recognized that violent injury prevention has the potential to completely prevent or alleviate the poor prognosis of some of the health problems faced by adults, or to delay their onset and/or reduce their severity. Therefore, in addition to striving to avoid violent incidents, medical and social welfare personnel should pay attention to the mental state of adult abuse victims, as well as the risk of poor prognosis (especially in PTSD, bipolar disorder, and manic disorder).

Future studies should investigate whether there have been any changes in the poor prognosis, including PTSD, bipolar disorder, manic disorder, etc., over the observation period from 2016 to 2022.

## Figures and Tables

**Figure 1 healthcare-10-01470-f001:**
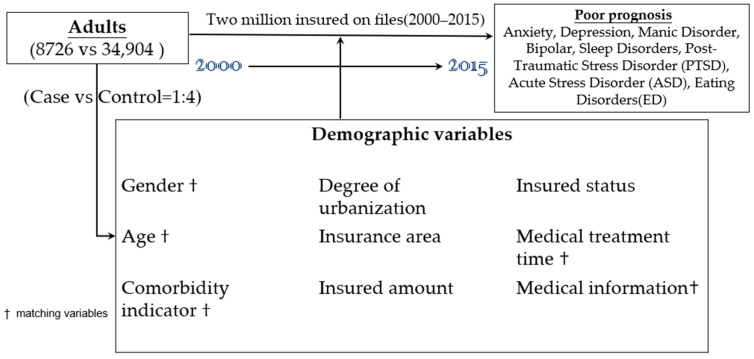
The flowchart of the study sample selection.

**Table 1 healthcare-10-01470-t001:** Basic demographic data of adult victims and control groups seeking medical treatment.

Demographic Variables		Adult Victims (*n* = 8726)		Control Group (*n* = 34,904)		*p*-Value
		*n*	*%*	*n*	*%*	
Gender	Female	2873	32.9	11,492	32.9	>0.999
	Male	5853	67.1	23,412	67.1	
Age	Mean ± SD, year	36.9	(12.4)	36.9	(12.2)	0.9279
Urbanization level	highly	1758	20.1	10,157	29.1	<0.001 **
	moderately	2513	28.8	11,255	32.2	
	new town	1934	22.2	6244	17.9	
	general townships	1340	15.4	4416	12.7	
	aging towns	716	8.2	1502	4.3	
	agricultural township	430	4.9	955	2.7	
	missing value	35	0.4	375	1.1	
Insurance Area	Taipei	2136	24.5	12,768	36.6	<0.001 **
	North District	1124	12.9	5081	14.6	
	Central District	2408	27.6	6238	17.9	
	South District	1034	11.8	4725	13.5	
	Kaohsiung and Pingtung	1737	19.9	5013	14.4	
	Eastern District	260	3.0	741	2.1	
	missing value	27	0.3	338	1.0	
Insured Amount	0–16,499 dollars	2870	32.9	5894	16.9	<0.001 **
(NTD)	16,500–20,999 dollars	2464	28.2	9434	27	
	21,000–30,299 dollars	2006	23.0	8238	23.6	
	≥30,300 dollars	1386	15.9	11,338	32.5	
Personal/Public Health Insurance	Public insurance	233	2.7	1997	5.7	<0.001 **
Dependent Occupation	Labor protection	4542	52.1	23,870	68.4	
	Farmer	841	9.6	2710	7.8	
	Member of Water Conservancy and Fisheries Association	317	3.6	866	2.5	
	Low-income Households	208	2.4	263	0.8	
	Community insured population	2557	29.3	4844	13.9	
	Other + missing values	28	0.3	354	1.0	

**: *p* < 0.05. HR = Hazard Ratio.

**Table 2 healthcare-10-01470-t002:** Comparison of the risk of poor prognosis between victims of violent abuse and control (Adults, Overall).

Poor Prognosis	Victims of Violence (*n* = 8726)	Not Victims (*n* = 34,904)	HR (95% CI)	*p*-Value
	Incidence (1/10^4^)	Incidence (1/10^4^)		
Psychiatric comorbidity				
Anxiety	216.0	142.0	1.50 (1.41–1.60)	<0.0001 **
Depression	138.0	61.2	2.27 (2.08–2.47)	<0.0001 **
Manic disorder	9.2	2.2	4.10 (2.84–5.94)	<0.0001 **
Bipolar disorder	34.0	7.9	4.44 (3.66–5.40)	<0.0001 **
Sleep disorder	338.1	264.1	1.29 (1.23–1.36)	<0.0001 **
Post-traumatic stress disorder	9.0	0.3	34.86 (15.94–76.2)	<0.0001 **
Acute stress disorder	21.3	9.0	2.52 (2.04–3.12)	<0.0001 **
Eating disorders	12.9	5.3	2.42 (1.84–3.19)	<0.0001 **

**: *p* < 0.05. HR = Hazard Ratio.

**Table 3 healthcare-10-01470-t003:** Comparison of the risk of poor prognosis between victims of violent abuse and controls (Adults, Males).

Poor Prognosis	Victims of Violence (*n* = 5853)				Not Victims (*n* = 23,412)				HR (95% CI)	*p*-Value
	*n*	%	Total Person-Years	Incidence (1/10^4^)	*n*	%	Total Person-Years	Incidence (1/10^4^)		
Psychiatric comorbidity										
Anxiety	832	14.2	45,697	182.1	2338	10.0	187,568	124.6	1.43 (1.32–1.55)	<0.0001 **
Depression	535	9.1	46,526	115.0	1029	4.4	191,689	53.7	2.18 (1.96–2.44)	<0.0001 **
Manic disorder	29	0.5	48,412	6.0	42	0.2	194,539	2.2	2.85 (1.74–4.67)	<0.0001 **
Bipolar disorder	132	2.3	48,053	27.5	145	0.6	194,270	7.5	3.81 (2.97–4.89)	<0.0001 **
Sleep disorder	1350	23.1	43,928	307.3	4362	18.6	182,982	238.4	1.30 (1.22–1.38)	<0.0001 **
Post-traumatic stress disorder	16	0.3	48,481	3.3	≤5			0.2	30.0 (6.86–131.18)	<0.0001 **
Acute stress disorder	71	1.2	48,276	14.7	144	0.6	194,281	7.4	2.09 (1.55–2.81)	<0.0001 **
Eating disorders	48	0.8	48,369	9.9	84	0.4	194,494	4.3	2.23 (1.54–3.23)	<0.0001 **

**: *p* < 0.05. HR = Hazard Ratio.

**Table 4 healthcare-10-01470-t004:** Comparison of the risk of poor prognosis between victims of violent abuse and controls (Adults, Females).

Poor Prognosis	Victims of Violence (*n* = 2873)				Not Victims (*n* = 11,492)				HR (95% CI)	*p*-Value
	*n*	%	Total Person-Years	Incidence (1/10^4^)	*n*	%	Total Person-Years	Incidence (1/10^4^)		
Psychiatric comorbidity										
Anxiety	567	19.7	19,057	297.5	1493	13.0	82,249	181.5	1.62 (1.46–1.79)	<0.0001 **
Depression	381	13.3	19,842	192.0	665	5.8	84,945	78.3	2.39 (2.10–2.73)	<0.0001 **
Manic disorder	35	1.2	21,342	16.4	19	0.2	86,673	2.2	6.71 (3.75–12.0)	<0.0001 **
Bipolar disorder	103	3.6	21,151	48.7	76	0.7	86,550	8.8	5.65 (4.12–7.73)	<0.0001 **
Sleep disorder	759	26.4	18,457	411.2	2574	22.4	79,691	323.0	1.28 (1.17–1.39)	<0.0001 **
Post-traumatic stress disorder	47	1.6	21,246	22.1	6	0.1	86,727	0.7	36.8 (14.62–92.61)	<0.0001 **
Acute stress disorder	77	2.7	21,188	36.3	109	0.9	86,505	12.6	3.09 (2.28–4.18)	<0.0001 **
Eating disorders	42	1.5	21,355	19.7	64	0.6	86,621	7.4	2.69 (1.79–4.03)	<0.0001 **

**: *p* < 0.05. HR = Hazard Ratio.

## Data Availability

Data are available from the National Health Insurance Research Database (NHIRD) published by the Taiwan National Health Insurance (NHI) Administration. Due to legal restrictions imposed by the government of Taiwan concerning the “Personal Information Protection Act”, data cannot be made publicly available. Requests for data can be sent as a formal proposal to the NHIRD (https://dep.mohw.gov.tw/DOS/lp-2506-113.html, accessed on 3 July 2022).

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
