# Peer review of "Adults Suffering from Violence Are at Risk of Poor Prognosis in Taiwan, 2000–2015"

_healthcare, 2022, doi:10.3390/healthcare10081470_

Round 1

Reviewer 1 Report

In this article, the authors used data of outpatient, emergency department and hospitalization from 2000 to 2015. The ICD code wqas 995.8 or E960-969.  They also analyzed a control group of non violent patients matched in a 1:4 ratio. During the 15 years period 8726 people suffered from violence and they experienced sleep disorders, anxiety and depression. The authors analyzed the risk of developing PTSD or bipolar disorder and manic disorder.  In conclusion, the authors found that people who suffered from violence had higher risk of poor prognosis. 

INTRODUCTION: very well written, maybe  increase the part dedicated to mental health and abuse (also early abuse if you can find literature). 

METHOD: Accurate. 

RESULTS: In my opinion, the tables are a little too confusing. 

DISCUSSION: Adults subjected to violence have higher risks of poor prognosis, in particular PTSD, bipolar disorder and manic disorder. 

OVERALL CONCLUSIONS: The article is well written accurate and uses data from the last fifteen years. It is not so innovative in terms of findings, but it is worth publishing in the present form.

Reviewer 2 Report

Needs major work

How is this study different from others and novel? 

There does not seem to be a comprehensive analyses of data-e.g. baseline measures were not controlled for?

The discussion is haphazard and needs to focus on implications for practice and research and what this study means?

Tables have terms that are confusing= violent injurer? means someone who injures others?

What are the limitations - expand on this?

What power analysis or sample size estimation methods were used?

How were type 1 and type 2 errors controlled for?

Reliability and validity of measures not discussed adequately and the effect on findings?

Most analyses is simplistic without using measures such as adjusted regressions and such?

Reviewer 3 Report

Title: Appropriate

Abstract: Good

Introduction: Satisfactory

Methods:

The authors should explain their data analysis approach and state the rationale for their chosen statistical tests.

The authors should state the meaning of the acronym GEE - ??? Generalized Estimating Equations

Overall the methods section appears confusing and needs to be made more understandable for ease of replication.

Results:

Please include the meaning of the acronym "HR" used severally in the results and discussion sections.

Table 1, 2, 3, & 4 footnotes are quite unclear. Kindly represent them in an easy-to-understand format.

Kindly consider changing the descriptive term "injurer" in your table titles. It doesn't aptly describe a victim or non-victim of violence.

Discussion: 

Please indicate the similarities in the prognosis of adults subjected to violence in the papers cited in lines 234 -238.

Also, It appears some of the papers cited, e.g., Hou et al., 2005 did not evaluate similar outcomes to the current study

Conclusion: ok

References:

Kindly provide a verifiable citation for reference 20 (preferably a digital object identifier - DOI)

Round 2

Reviewer 2 Report

Thank you